# Financial Inclusion, Fintech, and Income Inequality in Africa

**Biruk Birhanu Ashenafi** *[ID] and **Yan Dong**

Research Institute of Economics and Management, Southwestern University of Finance and Economics, Chengdu 611100, China
* Correspondence: birukb@smail.swufe.edu.cn

**Abstract:** Financial inclusion and Fintech have revolutionized the financial sector and fundamentally changed how we store, save, borrow, transfer, and invest money. This paper investigates the impact of financial inclusion and Fintech on income inequality using waves of survey data for 2011, 2014, and 2017 across 39 African countries. By using pooled ordinary least square and two-stage least square (2sls) estimation methods, we obtain three key findings. First, institutional factors such as political stability, control of corruption, and government effectiveness determine Fintech and financial inclusion. Second, Fintech encourages individuals to have a formal bank account, thereby promoting financial inclusion. Third, financial inclusion and Fintech exacerbate income inequality. The direct implication of our findings is that policymakers make tradeoffs whether they seek to achieve higher inclusion and Fintech or to reduce income inequality. We highlight that a pro-poor financial sector development is vital. Easing the bottleneck in obtaining loans, offering agriculture-based Fintech services, and improving digital literacy are important steps to gain the most out of inclusion and Fintech in reducing income inequality.

**Keywords:** Africa; Fintech; financial inclusion; income inequality; institutions

## 1. Introduction

A surge in income inequality has led to a renewed interest in understanding the driving forces of inequality. The potential determinants of income inequality could be economic, financial, demographic, institutional, technological, or policy factors [1]. However, there is a tendency in the mainstream literature to pay more attention to methodological issues in constructing top incomes or historical evolution [2–5]. Recently, however, key indicators from the financial sector have been attributed to the soaring income inequality in Africa. This paper extends the discussion on how financial sector development affects income inequality by focusing on an individual use of financial services. Research shows that the provision of financial infrastructure improves production and productivity [6] and eventually contributes to narrowing income inequality. In our approach, two concepts emerged: financial inclusion and financial technology (Fintech). We denote these aspects as the demand side of financial sector development.

We are motivated by the observation in Figure 1 that financial inclusion represented by account ownership and Fintech by digital payment exhibited a negative relationship with Gini disposable, which supports the current literature asserting inclusion and that Fintech reduces income inequality. Financial inclusion is one of Africa's great success stories over the past decade [7]. However, small-scale farmers are often excluded from formal financial services because of their low and seasonal income variation. Despite skyrocketing development in inclusion and Fintech, we argue that the key fundamentals that enable the financial sector development to play a role in alleviating allied societal problems such as income inequality are missing in Africa. In addition, people attain a broad range of financial services through various digital channels, including payments, credit, saving, and enabling banks to provide credit; however, the bottleneck remains unresolved. Individuals still suffered from accessing credit for start-ups, and uneven Fintech coverage contributed

to the soaring income inequality that places the continent as the second most unequal region next to Latin America [8].

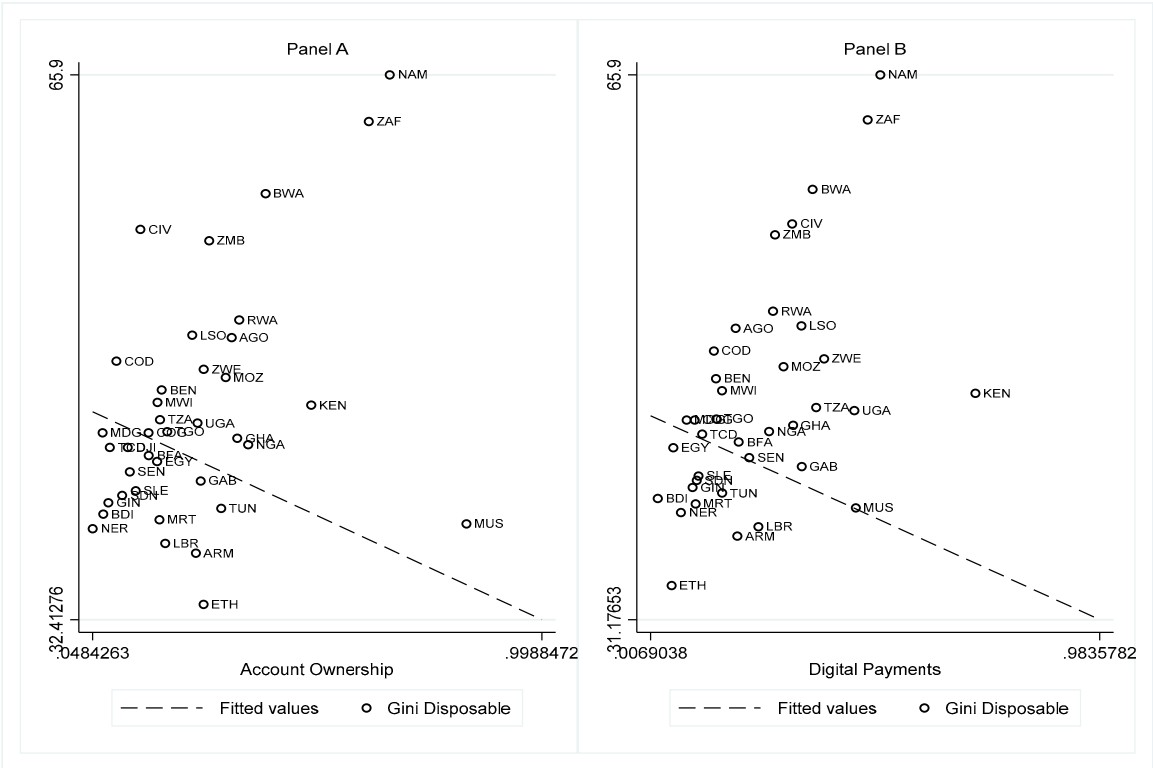

**Figure 1.** Relationship between financial inclusions, Fintech, and income inequality. Note: In panel A & B 39, countries are plotted using three waves of Global Findex survey in 2011, 2014, and 2017.

Therefore, we ask, can institutional quality affect financial inclusion and Fintech? Is Fintech important in achieving financial inclusion? How do inclusion and Fintech affect income inequality? A bulk of literature has asserted that financial inclusion reduces income inequality [9–11]. These papers claim that as more people from the low-income group access financial services, they will be able to move up the income ladder. Poor people can borrow and address their development needs when financial services are more accessible. However, a peculiar feature of these papers is that supply-side measures have been given due attention. Measurements such as ATM per 100,000 adults or bank per 100,000 adults have been used. A slightly different measure for financial inclusion credit to SMEs has been used by [10]. The current literature uses financial accessibility as the key indicator for inclusion. Our approach departs from this literature by focusing on the proportion of the population that uses these financial services. Another way inclusion can be promoted is through financial technology (Fintech). Ref. [12] find that financial inclusion is a key through which Fintech reduces income inequality.

The paper contributes to a growing body of literature on the finance–inequality nexus and presents evidence on the demand side of financial development. Our wisdom on how individuals' use of existing financial services affects macroeconomic fundamentals is limited. A key reason is measurement issues on inclusion and Fintech. We provide evidence by approaching financial inclusion and Fintech from the user's point of view. We frame our concept on how the development of an individual's use of financial services affects income inequality. In recent years, digital payment platforms have been mounting, yet their impact on income inequality is not tested. Thus, this paper followed the approach by [12] but extended the finding by incorporating institutional factors that shape financial inclusion and Fintech, and focusing on African countries with distinct features from the rest of the world.

To answer the main questions, we used three waves of survey data in 2011, 2014, and 2017 for 39 African countries from the Global Findex database. By employing the pooled ordinary least square (OLS) and two-stage least squared estimation (2sls) methods, we obtained three key findings. First, control of corruption, political stability, and government effectiveness are important institutional factors that determine the development of financial inclusion and Fintech. Second, Fintech positively and significantly affects financial inclusion. Third, financial inclusion and Fintech widen income inequality. Our findings challenge the current literature claiming both indicators negatively affect income inequality. Our result can be understood in light of the stage of financial development in most African countries. We highlight that the expansion of the sector needs to be checked in line with how it addresses the financial service needs of poor people, especially in rural areas.

## 2. Literature Review

Ref. [13] deconstruct financial inclusion as access and use of financial services by households. A broader deconstruction has also been given by [14] in constructing a financial inclusion index and used penetration, availability, and usage dimension. This deconstruction enables us to assess the channels in which inclusion affects income inequality. The first batch of literature views inclusion as expanding the accessibility of financial services and finds a negative relationship between financial inclusion and income inequality [10,11,15–17]. This stream of literature highlighted that greater financial inclusion helps reduce income inequality as more people in the lower income group have access to financial services and can move up the income ladder. Others focus on the individual use of financial services represented by account ownership, borrowing, and saving and obtain similar findings [12]. The key mechanism in both approaches is that removing financial constraints is likely to benefit peoples with lower income. Financial services trigger people to finance their needs from financial institutions and enable them to grow on the income ladder. It has been documented that access to bank accounts improves households' prospects for future income distribution [18].

The Fintech–inequality nexus is, however, scarcely researched. Mobile phone and internet connectivity are essential components in establishing the link. Ref. [19] find that enhanced internet penetration, fixed broadband subscriptions, and phone penetration have a net effect on reducing income inequality. Likewise, ref. [20] find that the interaction between mobile phones and the internet with primary school education narrows income inequality in Africa. Financial services that become more available through mobile payment replace development initiatives and allow developing countries to finance industrial and agricultural projects with local money [21]. The paper highlights that the mobile payment revolution has been taking place in Africa and other developing regions and shows that FinTech, such as M-Pesa, helps people to be "financially included" in Kenya.

On the other front, the role of institutions is undeniable in bringing the full effect of financial development. Governance factors catalyze efficient resource flow and enable the continent to use its abundant natural resources for societal development. For instance, corruption and political instability hinder the fruits of financial development not to be reaped by individuals. Corruption has a destructive impact on growth and business operations [22]. It affects the overall business environment and governance quality, and its impact is more pronounced in low-income countries. Corruption is associated with higher firm borrowing costs, lower stock valuation, worse corporate governance, bank stability, and risk for bank lending [23–25]. On the other hand, controlling corruption and its interaction with financial development has proven to reduce income inequality [26].

Pertinent to the rule of law, ref. [11] find that rule of law, per capita income, and demographic characteristics significantly affect financial inclusion. It is documented that the role of institutions in the fight to control corruption and interaction with domestic credit exhibit an inverted U-shape relationship with income inequality [27]. Furthermore, ref. [28] show that better governance and institution foster financial development in developing countries. Contrary to this, ref. [29] find that the effectiveness of legal institutions

does not promote stock market development in SSA. From the prevailing literature, we hypothesize that financial inclusion and Fintech significantly affect income inequality in Africa. Figure 2 schematically illustrates how the demand side financial development affects income inequality.

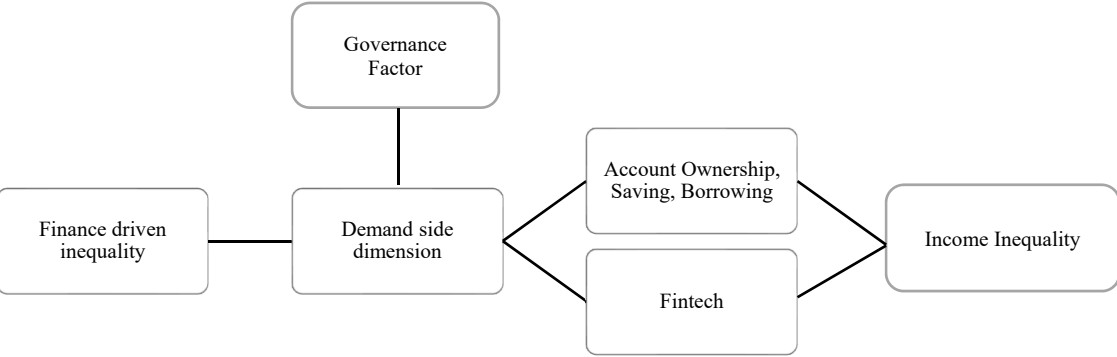

**Figure 2.** Schematic summary of theoretical relationship. Note: Omar and Inaba (2020) deconstruct financial inclusion in three dimensions, namely, penetration, availability, and usage of financial services, have been used. We further deconstruct into demand and supply-side developments and analyze the inclusion–Fintech inequality nexus by paying more attention to the demand-side indicators.

## 3. Materials and Methods

### 3.1. Model Specification

We used the following specification: Equation (1) specifies the role of Fintech on financial inclusion, and Equation (2) specifies the role of institutional quality on inclusion and Fintech.

$$Finance_{i,t} = \theta_0 + \theta_1 Fintech_{i,t} + \theta_n CV_{n,i,t} + \varepsilon_{i,t} \tag{1}$$

$$Finance_{i,t} = \theta_0 + \theta_1 Institutional\ Quality_{i,t} + \theta_n CV_{n,i,t} + \varepsilon_{i,t} \tag{2}$$

Then, we estimated the impact of financial inclusion and Fintech on income inequality:

$$logGini_{i,t} = \theta_0 + \theta_1 Financial\ Inclussion_{i,t} + \theta_2 FinTech_{i,t} + \theta_n CV_{n,i,t} + + \varepsilon_{i,t} \tag{3}$$

where the subscript $i, t,$ and $n$ indexes country, time, and the number of control variables, respectively. *Finance* measures *Financial Inclusion* and *Fintech*. We proxied financial inclusion by the percentage of adults who had an account at a financial institution and Fintech by digital payments made in the past year. *Institutional Quality* is measured by three governance indicators: political stability, control of corruption, and government effectiveness estimates. We used *logGini* to correct skewness in the distribution.

*CV* is a set of control variables. Various economic, institutional, financial, political, and demographic factors affect income inequality [30]. Economic growth affects wealth distribution [31]. Education complements the finance–inequality nexus [32]. It is worth mentioning that both predictors of financial development are linked to the level of education in a given country. Higher schooling paves the way for introducing and using financial products, including Fintech. A recent development on trade shows that it contributes to wealth and income inequality [30,33]. To control policy effects, we used government expenditure and inflation. The fiscal policy approach asserted that a high level of income inequality leads to a higher demand for redistribution [34]. The distributive effect of monetary policy and price stability also improves income inequality in a given country [35]. The demographic factor plays a significant role in studying income inequality [36]. These control variables were used to control cross-country variation.

The model was estimated using pooled ordinary least square regression (OLS). However, the estimates obtained from OLS might have been biased due to endogenous regressors. Thus, we cemented our finding using the instrumental variable approach and

estimated the relationship using the two-stage least square estimation (2sls) technique. Since our interest is investigating the effect of financial inclusion and Fintech on income inequality, we adopted the legal system as an instrument from the law and finance literature proposed by [37]. The legal origin instrument embeds the concept of rules protecting stakeholders and creditors, their origin, and the quality of their enforcement. Several works of literature use a legal origin dummy as an instrument for financial sector development [38,39]. Despite being widely adopted in the current literature, there is evidence that results for key indicators and specifications neither show common law to be consistently superior nor French civil law to be inferior to other legal families in generating strong financial development outcomes [40].

We argue that if legal institutions cannot protect customers or businesses, the percentage of adult people using the formal financial system declines. For this practical reason, we adopted the existing rule of law estimates instead of the legal origin dummy from [41]. Attributing the legal practices in the financial sector merely to the colonizer's law or the origin of their commercial law does less in settling contractual disputes. Instead, representing the current commercial or civil law enforcement practices could serve as a good candidate. The rule of law estimates captures perceptions of the extent to which agents have confidence in and abide by the rules of society, in particular, the quality of contract enforcement, property rights, the police, and the courts, as well as the likelihood of crime and violence.

### 3.2. Data

We obtained the data from different sources for the period of 2011–2017. The most widely used measure for country-level income inequality is the Gini index. The index measures the extent to which income among individuals or households deviates from a perfectly equal distribution. The Standardized World Income Inequality Database (SWIID) provides income inequality estimates across countries and over time using the Luxembourg Income Study [42]. It is a widely used dataset due to its broad coverage and allows comparison across countries. The dataset presents Gini disposable, an estimate of the Gini index of inequality in square root scale of household disposable income (post-tax, post-transfer), and Gini market income (pre-tax, pre-transfer).

Our main predictors were financial inclusion (financial institution account % age 15+) and Fintech (digital payment made in the past year % age 15+). We obtained the data from the World Bank Global Findex database, which provides more than 200 indicators on account ownership, payments, saving, credit, and financial resilience. Global Findex data were reported for all indicators by country, region, and income group for 2011, 2014, and 2017. Even though the survey has the fourth wave conducted for the year 2021, we could not proceed after the year 2017 for a major reason. Most African countries only have inequality data until 2017 and only a few until 2019. Therefore, pushing the year until 2021 would lead us to suffer from missing data. Given that research in the area is limited in Africa, we were forced to document the nexus until 2017. The other vectors were institutional factors from the World Bank Worldwide Governance Indicators (WGI), constructed by Kaufmann et al. (2010b). We used political stability, control of corruption, and government effectiveness to proxy institutional quality. All the control variables were obtained from the World Bank World Development Indicators (WDI). Table 1, summarizes variable definition and sources.

From the summary statistics in Table 2, the percentage of adults who have an account in a formal financial institution is 26.4%, lower than the world average of 53.6%. These statistics suggest that African countries are far behind in getting people into the formal financial sector. This performance, among others, is responsible for the low level of savings, at 11.8%, and borrowings, at 6.5%, which are vital to the financial sector's resilience. It is worth mentioning that without an inclusive financial system, meeting the capital need of the continent is challenging. Meanwhile, recent years have shown tremendous development in digital payment systems. The continent only has 24.3% of its adult population using digital

payment systems, which is also far lower than the world average of 43.4%. Furthermore, the pairwise correlation shows a positive and significant relationship between account ownership, Fintech, savings, and income inequality. We formalized the statistics in the latter section by controlling key country-level indicators.

**Table 1.** Variable Description.

| Variable | Short Definition | Source |
|---|---|---|
| Gini | Gini disposable | The Standardized World Income Inequality Database (SWIID) |
| Account Ownership | Financial institution account (% age 15+) | Global Findex |
| Fintech | Made digital payments in the past year (% age 15+) | Global Findex |
| Savings | Saved at a financial institution (% age 15+) | Global Findex |
| Borrowing | Borrowed from a financial institution (% age 15+) | Global Findex |
| GDP Growth | GDP per capita growth (annual %) | World Development Indicators |
| Schooling | School enrollment, primary and secondary (gross), gender parity index (GPI) | World Development Indicators |
| Trade | Trade (% of GDP) | World Development Indicators |
| Gov't Expenditure | General government final consumption expenditure (% of GDP) | World Development Indicators |
| Inflation | Inflation, GDP deflator (annual %) | World Development Indicators |
| Population | Population growth (annual %) | World Development Indicators |
| Institutional Quality | Control of corruption, political stability, government effectiveness | World Governance Indicator (WGI) |

Note: The table presents the variables used in the paper.

**Table 2.** Summary statistics.

| | Variable | Obs | Mean | Std. Dev. | Min | Max |
|---|---|---|---|---|---|---|
| [26] | Gini | 91 | 45.178 | 7.221 | 33.3 | 65.9 |
| [27] | Account Owners | 102 | 0.264 | 0.188 | 0.015 | 0.895 |
| [29] | Fintech | 69 | 0.243 | 0.171 | 0.022 | 0.764 |
| [2] | Savings | 102 | 0.118 | 0.084 | 0.006 | 0.355 |
| [19] | Borrowing | 102 | 0.065 | 0.045 | 0.013 | 0.284 |
| [3] | GDP Growth | 101 | 2.505 | 3.689 | −6.809 | 24.976 |
| [43] | Schooling | 61 | 0.937 | 0.106 | 0.630 | 1.071 |
| [23] | Trade | 96 | 69.531 | 26.963 | 1.377 | 149.01 |
| [44] | Gov't Expen | 94 | 14.584 | 6.324 | 4.325 | 39.690 |
| [22] | Inflation | 101 | 7.755 | 9.699 | −11.876 | 60.987 |
| [38] | Population | 102 | 2.440 | 0.878 | −0.027 | 3.899 |

| | [26] | [27] | [29] | [2] | [19] | [3] | [43] | [23] | [44] | [22] | [38] |
|---|---|---|---|---|---|---|---|---|---|---|---|
| [26] | 1 | | | | | | | | | | |
| [27] | 0.388 * | 1 | | | | | | | | | |
| [29] | 0.478 * | 0.705 * | 1 | | | | | | | | |
| [2] | 0.385 * | 0.877 * | 0.621 * | 1 | | | | | | | |
| [19] | −0.045 | 0.551 * | 0.455 * | 0.457 * | 1 | | | | | | |
| [3] | −0.104 | 0.159 | −0.102 | 0.122 | 0.101 | 1 | | | | | |
| [43] | 0.023 | 0.368 * | 0.311 * | 0.291* | 0.421 * | 0.161 | 1 | | | | |
| [23] | 0.147 | 0.177 | −0.031 | 0.056 | 0.033 | 0.024 | 0.023 | 1 | | | |
| [44] | 0.396 * | 0.200 | 0.144 | 0.108 | −0.086 | −0.099 | 0.273 * | 0.461 * | 1 | | |
| [22] | 0.025 | −0.041 | −0.025 | −0.034 | −0.053 | 0.013 | −0.013 | −0.111 | −0.155 | 1 | |
| [38] | −0.085 | −0.531 * | −0.271 * | −0.303 * | −0.489 * | −0.216 * | −0.511 * | −0.397 * | −0.352 * | 0.088 | 1 |

Note: * shows $p < 0.05$. The summary statistics is conducted for 39 countries. Few countries only have one wave of data. Moreover, Fintech, which represents a digital payment made last year, is surveyed for 2014 and 2017.

## 4. Empirical Results

The first exercise presents the impact of institutional quality on financial inclusion and Fintech. In the past decade, the development of account ownership was stagnant at 26%. Likewise, digital financial platforms grew by less than 1 percentage point from 23.8% in 2011 to 24.7% in 2017. We claim that institutional factors and digitalization performance shape the current development. Our institutional factors include political stability, control of corruption, and government effectiveness, which shape the development over time in Table 3. The distribution of institutional quality estimates ranges from −2.5 (the lowest) to +2.5 (the highest). The average estimates related to political stability, control of corruption, and government effectiveness were at −0.58, −0.61, and −0.67, respectively, putting the continent in a lower position to realize conducive conditions for development. A quality institutional environment is vital in explaining financial development [45]. Institutional factors affect some aspects of financial development by creating trust. Individuals should trust the existing institutional effectiveness in using financial services. We show that while political stability, control of corruption, and government effectiveness are key determinants of financial inclusion, the latter two factors significantly affect Fintech.

**Table 3.** The role of institutional quality on financial inclusion and fintech.

| Dependent Variables | | Financial Inclusion | Fintech | | Financial Inclusion | Fintech | | Financial Inclusion | Fintech |
|---|---|---|---|---|---|---|---|---|---|
| | | I. | II. | | III. | IV. | | V. | VI. |
| Indep Var | Political Stability | 0.076 * | 0.047 | Control of Corruption | 0.176 *** | 0.120 * | Government Effectiveness | 0.230 *** | 0.138 ** |
| | | (0.032) | (0.037) | | (0.046) | (0.048) | | (0.036) | (0.043) |
| Observation | | 67 | 44 | | 67 | 44 | | 67 | 44 |
| R-Squared | | 0.36 | 0.11 | | 0.434 | 0.205 | | 0.588 | 0.277 |

Note: Standard errors in parentheses. * $p < 0.05$, ** $p < 0.01$, *** $p < 0.001$. We control GDP per capita in all specifications.

Fintech is the other factor that plays a role in facilitating inclusion. In many parts of the world, the launch and growth of digital financial services have led to an unprecedented increase in people enjoying access to formal financial services [7]. Fintech, in the form of mobile money and agent banking, offers affordable, instant, and reliable transactions, savings, and credit in rural villages where banks are rarely operating. This development of banking services at the fingertips indirectly incentivizes people to have a bank account [46]. Digital payment platforms are becoming vital in Africa. Although the development is at its early stage, a few countries have managed to perform higher than the global average. Africa's top three digital economies include Kenya, Namibia, and South Africa, with approximately 71.3%, 50%, and 47.8% of transactions made through digital payment platforms. Notably, the East African countries are becoming Fintech hubs in recent years. Contrary to this, Burundi, Ethiopia, and Egypt are the lowest performers on digital payment platforms, with only 2.%, 5.2%, and 5.6% of transactions made by digital payment platforms.

Our finding shows that Fintech has a colossal role in achieving financial inclusion by easing the opening of a formal financial account in Table 4. This implies the digital transformation approach pursued by a few African countries has the potential to achieve financial inclusion. The finding corroborates [12] that Fintech is an enabler for financial inclusion. Furthermore, the benefit extends to mobilizing savings from account holders. The use of Fintech attracts a considerable amount of money circulating outside the banking system. Given that Fintech increases saving, the finding lends vital implications on achieving a holistic financial development approach to facilitate financial intermediation.

**Table 4.** The impact of fintech on financial inclusion, savings, and borrowing.

| Dependent Variables | Financial Inclusion | Savings | Borrowing | Financial Inclusion | Savings | Borrowing |
|---|---|---|---|---|---|---|
| | OLS | | | 2sls | | |
| | I. | II. | III. | IV. | V. | VI. |
| Fintech | 0.817 *** | 0.331 *** | 0.0226 | 1.455 *** | 0.655 *** | 0.0487 |
| | (0.132) | (0.079) | (0.039) | (0.338) | (0.192) | (0.074) |
| GDP PC Growth | −0.002 | −0.002 | 0.004 * | - | - | - |
| | (0.007) | (0.004) | (0.002) | | | |
| Schooling | −0.138 | 0.0155 | 0.0234 | −0.24 | −0.039 | 0.037 |
| | −0.197 | −0.118 | −0.0596 | −0.231 | (0.131) | (0.057) |
| Trade | −0.005 | −0.003 | −0.001 | −0.001 | −0.001 | −0.0006 |
| | (0.008) | (0.005) | (0.002) | (0.001) | (0.005) | (0.002) |
| Government Expen | −0.004 | −0.009 | −0.001 | 0.007 | −0.005 | −0.001 * |
| | (0.003) | (0.002) | (0.009) | (0.003) | (0.002) | (0.008) |
| Inflation | −0.001 | −0.003 | −0.005 | 0.005 | 0.0005 | −0.007 |
| | (0.002) | (0.001) | (0.005) | (0.002) | (0.001) | (0.005) |
| Population Growth | −0.111 *** | −0.0231 | −0.0411 *** | −0.067 | −0.008 | −0.039 *** |
| | (0.027) | (0.016) | (0.008) | (0.037) | (0.021) | (0.008) |
| Constant | 0.539 * | 0.122 | 0.156 * | 0.36 | 0.0312 | 0.149 * |
| | (0.254) | (0.152) | (0.076) | (0.308) | (0.174) | (0.076) |
| N | 44 | 44 | 44 | 44 | 44 | 44 |
| adj. R-sq | 0.703 | 0.389 | 0.584 | 0.521 | 0.122 | 0.544 |
| Sargan Statistics | - | - | - | 0.6853 | 0.566 | 0.0621 |

Note: Standard errors in parentheses. * $p < 0.05$, ** $p < 0.01$, *** $p < 0.001$. Columns IV-VI, we use the rule of law estimates as an instrument. We control GDP per capita growth, schooling, government expenditure, inflation, and population growth. Of the 39 countries surveyed, only 25 have Fintech data recorded in two waves.

We now turn to assess how financial inclusion affects income inequality. As we have discussed previously, even though the growth of account ownership is sluggish, the digital platform brought additional benefits for inclusion and mobilized savings. Our finding shows that financial inclusion and Fintech positively affect Gini disposable in Table 5.

Higher account ownership implies more people are banked and mobilize higher capital that has been circulated outside the banking system. This development, in turn, gives banks more space to supply credit. There is a growing consensus that lack of access to finance adversely affects economic growth, poverty alleviation, and human development [47]. The finding shows that structural problems weaken the impact of inclusion on alleviating inequality problems. We claim that the bank's behavior could drive the result. Most African countries have experienced financial friction, information asymmetry, and excessive regulation. In addition to these, per capita income is the lowest globally. This myriad of structural problems pushes financial institutions to choose low-risk investment opportunities owned and operated by big companies or governmental bodies. As a result, when financial development enhances the services to those already accessing the financial system (high-income earners), it widens income inequality [48]. Through inclusion mobilizing, capital alone does not solve problems associated with credit rationing that small businesses face. When lending is highly collateralized, it narrows the space for poor people to access finance and widens the income gap. Apart from that, financial literacy, which is used in financial decision-making [49], is lacking in many parts of rural areas.

Likewise, a rise in the use of Fintech increases Gini disposable. The development of Fintech enables people to access loans easily and allows purchasing agricultural inputs with less financial friction [50]. Two potential mechanisms are worth mentioning that explain why Fintech exacerbates income inequality. First, we show that Fintech is not correlated to GDP per capita income in Table 2. Under this circumstance, the mere usage of a digital payment system could help the banking system raise capital, as we have discussed earlier, but it has no effect on improving household income. In a situation where income is not improved for most people, income distribution cannot be improved. With low per capita income, a rising Fintech may help account for ownership. However, the amount of

capital mobilized and redirected to investment opportunities is insufficient to negatively impact inequality.

**Table 5.** Financial inclusion, Fintech, and income inequality.

| | **Dependent Variable: log[Gini Disposable]** | | | | | | | |
| | **OLS** | | | | | **2sls** | | |
| | **I.** | **II.** | **III.** | **IV.** | **V.** | **VI.** | **VII.** | **VIII.** |
|---|---|---|---|---|---|---|---|---|
| Financial Inclusion | 0.284 *** | | 0.332 *** | | −0.043 | 0.336 * | | −0.0153 |
| | (0.078) | | (0.095) | | (0.189) | (0.142) | | (0.294) |
| Fintech | | 0.440 *** | | 0.578 *** | 0.616 ** | | 0.746 ** | 0.755 |
| | | (0.107) | | (0.140) | (0.219) | | (0.275) | (0.492) |
| GDP PC Growth | | | −0.001 | −0.021 | −0.021 | | | |
| | | | (0.006) | (0.011) | (0.012) | | | |
| Schooling | | | −0.088 | −0.231 | −0.242 | −0.093 | −0.229 | −0.248 |
| | | | (0.190) | (0.239) | (0.248) | (0.176) | (0.295) | (0.252) |
| Trade | | | −0.007 | −0.008 | −0.008 | −0.007 | −0.005 | −0.005 |
| | | | (0.007) | (0.001) | (0.001) | (0.006) | (0.001) | (0.001) |
| Government Expen | | | 0.006 * | 0.002 | 0.002 | 0.007 ** | 0.005 | 0.005 |
| | | | (0.003) | (0.004) | (0.004) | (0.002) | (0.003) | (0.003) |
| Inflation | | | 0.004 | −0.005 | −0.005 | 0.042 | 0.043 | 0.041 |
| | | | (0.002) | (0.002) | (0.003) | (0.027) | (0.035) | (0.037) |
| Population Growth | | | 0.045 | 0.020 | 0.015 | 0.004 | 0.001 | 0.001 |
| | | | (0.025) | (0.031) | (0.037) | (0.002) | (0.002) | (0.003) |
| Constant | | | 3.633 *** | 3.896 *** | 3.926 *** | 3.628 *** | 3.688 *** | 3.712 *** |
| | | | (0.243) | (0.310) | (0.341) | (0.234) | (0.319) | (0.319) |
| N | 91 | 58 | 59 | 36 | 36 | 59 | 35 | 36 |
| adj. R-sq | 0.119 | 0.219 | 0.148 | 0.311 | 0.287 | 0.164 | 0.191 | 0.19 |
| Sargan Statistics | - | - | - | - | - | 0.8099 | 0.0961 | 0.0637 |

Note: Standard errors in parentheses. * $p < 0.05$, ** $p < 0.01$, *** $p < 0.001$. Columns I–V estimation is based on OLS, and column VI–VIII uses 2sls using the rule of law estimates as an instrument. We control GDP per capita growth, schooling, government expenditure, inflation, and population growth. Of the 39 countries surveyed, only 25 countries have Fintech data recorded in two waves.

The second mechanism is related to platform ownership. We show a positive correlation between schooling and Fintech in Table 2, and rich entrepreneurs usually own payment platforms. This relative position enables platform developers to make money from subscriptions, third parties, and advertisements and raise short-term capital for business expansion. When this phenomenon continues, it generates a significant sum of capital for the developers and enables them to become richer. The net effect of Fintech eventually benefits the rich more than the poor and exacerbates income inequality.

There are also associated risks with Fintech. With low schooling, the rapid expansion of digital payments in the continent may not consider consumer protection and financial credibility. Ref. [51] demonstrates that—the advantages notwithstanding—Fintech solutions leave the door open to many risks that may hamper consumer protection and financial stability. Therefore, continental Fintech development needs to improve product features and shorten their delivery channels, thereby enhancing the convenience of accessing credit [52].

In a nutshell, the empirical exercise portrays how institutional factors are vital determinants of financial inclusion and Fintech. Moreover, Fintech enhances account ownership and paves the way to mobilize funds. Furthermore, financial inclusion and Fintech are proven to exacerbate income inequality.

### 5. Concluding Remark

Our empirical findings suggest that institutional factors are key determinants of financial inclusion and Fintech, Fintech facilitates financial inclusion, and financial inclusion and Fintech exacerbates income inequality. The results are robust for alternative estimation. The evidence presented in this paper departs from the current literature, potentially attributed to the early stage of financial sector development.

Two empirical implications were obtained from our exercise. First, a tradeoff exists between financial inclusion, Fintech, and income inequality. Financial institutions serve as a good instrument to narrow income inequality. Finance, if appropriately managed, can increase per capita income, alleviate poverty, and narrow inequality. Within this context, a workable solution to settle the tradeoff between financial development and income inequality is mobilizing the required capital by getting more people banked. This measure increases funds available in the banking system so that capital constraints for small businesses will be lessened. However, our evidence shows that higher account ownership and digital payment systems exacerbated income inequality. The direct implication is thus that the current capital mobilized through the system is not enough to fulfill the capital needs of small businesses. By expanding inclusion and Fintech, banks can gain sufficient funds to tackle the credit rationing problem. Therefore, easing the bottleneck in obtaining loans, offering agriculture-based Fintech services, and improving digital literacy are important steps to gain the most out of inclusion and Fintech in reducing income inequality.

The second implication is pertinent to the development of Fintech. The principal factor that has driven the growth of Fintech is the shallow bank distribution. In most parts of Africa, it is still difficult and sometimes even impossible to transfer money and pay bills. A workable solution to benefit from Fintech is adopting a telecom-led regulatory model. In this framework, as primary service providers, telecom companies ensure that services satisfy poor people's needs are developed and configured. For instance, M-Pesa is the most popular mobile-based money transfer service in Kenya and Tanzania, owned by Vodafone. The service enables customers to transfer, deposit, and withdraw money. The platform positively impacted the economies of Kenya and Tanzania by easing transactions, increasing per capita income, and solving temporary liquidity problems of households. It has also increased financial resilience by enabling higher savings. Therefore, improving digital literacy, strengthening the regulatory framework to minimize associated Fintech risks, and configuring financial services that help the poor and rural areas should receive considerable attention.

Our results should be treated with caution. Given the limited recent data available for income inequality, financial inclusion, and Fintech, the findings constitute an initial point of analysis of a topic that has been widely disregarded in the literature, especially in Africa. Future research should use longer-term data with more depth and coverage.

**Author Contributions:** Both authors contributed their share during the data management, analysis, and writing stage. All authors have read and agreed to the published version of the manuscript.

**Funding:** This research received no external funding.

**Institutional Review Board Statement:** Not applicable.

**Informed Consent Statement:** Not applicable.

**Data Availability Statement:** The datasets generated and analyzed during the current study are available from the Standardized World Income Inequality Database (SWIID) (Solt, 2019), version 9.1, the Global Findex, Kaufmann et al. (2010b), and World Development Indicator (WDI).

**Conflicts of Interest:** There is no conflict of interest, financial or non-financial.

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
