# Peer review of "Financial Inclusion, Fintech, and Income Inequality in Africa"

_fintech, doi:10.3390/fintech1040028_

Round 1

Reviewer 1 Report

Dear authors,

The article at the moment is not ready for publication. The main problem is old data for such a topic. The survey must be renewed. The literature must be updated as well. 

Author Response

  1. The main problem is old data for such a topic. The survey must be renewed.

Response: Agree. Your concern is reasonable; we have used a bit older data. As you can see, our dependent variable is obtained from SWIID in which the latest version released on June 23rd, 2022 (version 9.3) has a record only up to 2019. The majority of African countries only have till 2017. Likewise, our main predictors are obtained from the Global Findex database, the only reliable demand-side survey data we can use for African countries, especially Fintech. Until we wrote the article, 3 waves of survey data were released, but recently the 4th wave came out which makes the survey extends to 2021 (i.e., 2011, 2014, 2017, and 2021).

Given that the SWIID has no data points after 2017 for most African countries, we decided not to go after the year 2017. Although we find observations for a few countries till 2019, we conducted our analysis till 2017 to minimize missing data points. Despite the data limitation, our motive in conducting this empirical work is to document the evidence so that other researchers may push the knowledge frontier in the area in Africa.

  1. The literature must be updated as well.

Response: Agree. We made significant improvements on our literature in this version.

Reviewer 2 Report

The paper is scientifically sound and relevant to the aim and scope of the Journal. The methodological approach is aligned with the objectives and dataset. The empirical part of the paper is well-structured and represents important findings on links between financial awareness and connected indicators with income inequality. However, there are some important drawbacks of the research that should be corrected before publication:

(1) the authors investigate the issues of financial inclusion and Fintech influence on income inequality using the data of surveys "for 2011, 2014, and 2017 for 39 African countries from the Global Findex database". This information is obviously outdated for the findings and conclusions in 2022, especially if one can try to use the results for policymaking. So, authors should add data for analysis if possible or explain the usage of this statistical dataset. If recent data relevant to the field of study are unavailable, it should be explained. I find it possible to use the authors' approach as an example of the development of tools for the analysis of general links nature, i.e., for the purpose of methodology development in the field, but not for the current conclusions;

(2) literature review is too generalized in some parts of the article. It causes the impression of insufficient familiarity with the peculiarities of the previous researches. See, for example: "A growing body of empirical literature shows financial inclusion narrows income inequality (Beck et al., 2007; 3 Park & Mercado, 2015; Kim, 2016; Martínez Turégano & García Herrero, 2018; Park & Mercado, 2018; Neaime & Gaysset, 2018; Le et al., 2019; Huang & Zhang, 2020; Omar & Inaba, 2020; Demir et al., 2020; Khan et al., 2021)". It is totally inappropriate to cite 10 sources in a general manner without any explanation of scientific appropriateness for the own study, at least, briefly.

(3) It is surprising that such a thriving area if research is analyzed by authors using mainly outdated sources. Many of them were published in 1997, 1998, 2001, 2006.... Among these sources, only a few belong to the fundamental works in the field, like studies of Kuznets and Atkinson. the reference list and literature review sections should be updated using more recent studies. In light of the authors' research, these references can be beneficial: https://doi.org/10.1007/978-3-319-74216-8_10 ; doi:10.14254/2071-8330.2020/13- 4/21;  doi:10.14254/2071-789X.2020/13-3/9

(4) It would be appropriate to indicate the limitations of the research in the conclusion section, especially considering note 1 on the dataset. 

Author Response

(1) the authors investigate the issues of financial inclusion and Fintech influence on income inequality using the data of surveys "for 2011, 2014, and 2017 for 39 African countries from the Global Findex database". This information is obviously outdated for the findings and conclusions in 2022, especially if one can try to use the results for policymaking. So, authors should add data for analysis if possible or explain the usage of this statistical dataset. If recent data relevant to the field of study are unavailable, it should be explained. I find it possible to use the authors' approach as an example of the development of tools for the analysis of general links nature, i.e., for the purpose of methodology development in the field, but not for the current conclusions;

Response: Agree. Your concern is reasonable; we have used a bit older data. As you can see, our dependent variable is obtained from SWIID in which the latest version released on June 23rd, 2022 (version 9.3) has a record only up to 2019. The majority of African countries only have till 2017. Likewise, our main predictors are obtained from the Global Findex database, the only reliable demand-side survey data we can use for African countries, especially Fintech. Until we wrote the article, 3 waves of survey data were released, but recently the 4th wave came out which makes the survey extends to 2021 (i.e., 2011, 2014, 2017, and 2021).

Given that the SWIID has no data points after 2017 for most African countries, we decided not to go after the year 2017. Although we find observations for a few countries till 2019, we conducted our analysis till 2017 to minimize missing data points. Despite the data limitation, our motive in conducting this empirical work is to document the evidence so that other researchers may push the knowledge frontier in the area in Africa.

 (2) literature review is too generalized in some parts of the article. It causes the impression of insufficient familiarity with the peculiarities of the previous researches. See, for example: "A growing body of empirical literature shows financial inclusion narrows income inequality (Beck et al., 2007; 3 Park & Mercado, 2015; Kim, 2016; Martínez Turégano & García Herrero, 2018; Park & Mercado, 2018; Neaime & Gaysset, 2018; Le et al., 2019; Huang & Zhang, 2020; Omar & Inaba, 2020; Demir et al., 2020; Khan et al., 2021)". It is totally inappropriate to cite 10 sources in a general manner without any explanation of scientific appropriateness for the own study, at least, briefly.

Response: Agree. Thank you for pointing that out. In this version, we tried to give the review a shape following two decompositions. We also removed papers with broader implications than the topic raised in this paper. However, we kept few papers that we thing are still relevant.

(3) It is surprising that such a thriving area if research is analyzed by authors using mainly outdated sources. Many of them were published in 1997, 1998, 2001, 2006.... Among these sources, only a few belong to the fundamental works in the field, like studies of Kuznets and Atkinson. the reference list and literature review sections should be updated using more recent studies. In light of the authors' research, these references can be beneficial: https://doi.org/10.1007/978-3-319-74216-8_10 ; doi:10.14254/2071-8330.2020/13- 4/21;  doi:10.14254/2071-789X.2020/13-3/9

Response: Agree. We incorporated the comment and suggestions. In a few places, we still introduce contemporary works to shed some light to the readers in understanding the issue.

(4) It would be appropriate to indicate the limitations of the research in the conclusion section, especially considering note 1 on the dataset.

Response: Agree. We incorporated your comment and highlighted the limitation in the conclusion part

Reviewer 3 Report

I review the paper entitled “Financial Inclusion, Fintech, and Income Inequality in Africa” deeply. Before acceptance, please revise the paper according to my comments and suggestions.

1.      Please clearly indicate the theoretical background between financial inclusion and income inequalities. It would be betters if the authors make section-2 for theoretical background. The theoretical background would only show the pure channels and mechanism by which Financial Inclusion, Fintech, and Income Inequality effects each other’s.

2.      Please briefly explain the novelty by comparing another studies.

3.      Please includes some recent literature for example;

Financial infrastructure—total factor productivity (TFP) nexus within the purview of FDI outflow, trade openness, innovation, human capital and institutional quality: Evidence from BRICS economies. Applied Economics, 10.1080/00036846.2022.209433

4.      Please before going for the main regression i.e. OLS it is necessary that authors should apply some pre-test for example, Cross-sectional dependency test and Second generation Panel unit root test.

5.      Conclusion should be match with the results please.

Good Luck!

Author Response

  1. Please clearly indicate the theoretical background between financial inclusion and income inequalities. It would be betters if the authors make section-2 for theoretical background. The theoretical background would only show the pure channels and mechanism by which Financial Inclusion, Fintech, and Income Inequality effects each other’s.

Response: Agree. We tried to provide a schematic summary of on the channels on which inclusion and Fintech affect inequality in the literature part.

  1. Please briefly explain the novelty by comparing another studies.

Response: Agree. We add the paper contribution in the 3rd and 4th paragraphs.

  1. Please includes some recent literature for example;

Response: Agree. We improved our reference lists upon your recommendation. We also thank you for the suggested article.

  1. Please before going for the main regression i.e. OLS it is necessary that authors should apply some pre-test for example, Cross-sectional dependency test and Second generation Panel unit root test.

Response: We argue that these pretests may not be applied to our dataset for the following reasons: First, as you can see, our data is not collected continually. The survey is conducted in 2011, 2014, and 2017. Given that our panels have a short time period, the model may not suffer from crossectional correlated error terms. The same logic works for the panel unit root test. In a multiple regression analysis, a minimum of 10 data points for each vector is required (in our case, we only have 3 observations in each country and variable). Few references (1) Westland, J. Christopher (2010). "Lower bounds on sample size in structural equation modeling". Electron. Comm. Res. Appl. 9 (6): 476–487. (2) Nunnally, J. C. (1967). "Psychometric Theory". McGraw-Hill, New York: 355. (3) Yamane, Taro. 1967. Statistics: An Introductory Analysis, 2nd Ed., New York: Harper and Row

  1. Conclusion should be match with the results please.

Response: Agree. We made a few amendments to this version

Round 2

Reviewer 2 Report

Dear authors, the corrections fulfilled are essential mostly. The justification of the database you included in the article is appropriate and it suits to illustrate how the proposed method can be used in general, not only for the given range of data. 

However, I still find it necessary to consider some comments before publishing. Answering my 3rd comment you write that it is perceived by you and changes are done. But, you still use to narrowed background considering the corruption influence and shadow economy respectively. In your text, you write "For instance, corruption and political instability hinder the fruits of financial development not to be reaped by individuals". It fact, the corruption and shadow economy derived from it, hinder the development of social groups, deepening inequality as illustrated in https://link.springer.com/chapter/10.1007/978-3-319-74216-8_10 and many other sources in the field. Besides, it is directly related to the aspects of inequality highlighted by your team. It is inappropriate to justify this aspect by referring to sources published in 2006, 2014... It does not mean you should remove them if you really find them important, but including some new ones will be positive.

Please, check added fragments of text carefully. There are some grammar inconsistencies like " controlling corruption and its interaction with financial development has proven to reduce income inequality".

I hope these comments are useful for you and help to improve the quality of the manuscript before the final decision.

Reviewer 3 Report

I accept the paper in the current forr